# Targeted therapies for previously treated advanced or metastatic renal cell carcinoma: systematic review and network meta-analysis

Charlotta Karner, Kayleigh Kew, Victoria Wakefield, Natalie Masento, Steven J Edwards

British Medical Journal Technology Assessment Group (BMJ-TAG), BMA House, London, UK

**Correspondence to**
Dr Steven J Edwards;
sedwards@bmj.com

## ABSTRACT

**Objective** To compare the effectiveness and safety of treatments for advanced or metastatic renal cell carcinoma (amRCC) after treatment with vascular endothelial growth factor (VEGF)-targeted treatment.

**Design** Systematic review and network meta-analysis of randomised controlled trials (RCTs) and comparative observational studies. MEDLINE, EMBASE and Cochrane Library were searched up to January 2018.

**Participants** People with amRCC requiring treatment after VEGF-targeted treatment.

**Interventions** Axitinib, cabozantinib, everolimus, lenvatinib with everolimus, nivolumab, sorafenib and best supportive care (BSC).

**Outcomes** Primary outcomes were overall survival (OS) and progression-free survival (PFS); secondary outcomes were objective response rate (ORR), adverse events, and health-related quality of life (HRQoL).

**Results** Twelve studies were included (n=5144): five RCTs and seven observational studies. Lenvatinib with everolimus significantly increased OS and PFS over everolimus (HR 0.61, 95% Credible Interval [95%CrI]: 0.36 to 0.96 and 0.47, 95%CrI: 0.26 to 0.77, respectively) as did cabozantinib (HR 0.66, 95%CrI: 0.53 to 0.82 and 0.51, 95%CrI: 0.41 to 0.63, respectively). This remained the case when observational evidence was included. Nivolumab also significantly improved OS versus everolimus (HR 0.74, 95%CrI: 0.57 to 0.93). OS sensitivity analysis, including observational studies, indicates everolimus being more effective than axitinib and sorafenib. However, inconsistency was identified in the OS sensitivity analysis. PFS sensitivity analysis suggests axitinib is more effective than everolimus, which may be more effective than sorafenib. The results for ORR supported the OS and PFS analyses. Nivolumab is associated with fewer grade 3 or grade 4 adverse events than lenvatinib with everolimus or cabozantinib. HRQoL could not be analysed due to differences in tools used.

**Conclusions** Lenvatinib with everolimus, cabozantinib and nivolumab are effective in prolonging the survival for people with amRCC subsequent to VEGF-targeted treatment, but there is considerable uncertainty about how they compare to each other and how much better they are than axitinib and sorafenib.

**PROSPERO registration number** CRD42017071540.

### Strengths and limitations of this study

► This review is highly relevant and timely as it includes all recently approved treatments and focuses on the effectiveness of these treatments when used after first-line VEGF-targeted tyrosine kinase inhibitor (TKI) treatment, as recommended in European clinical guidelines.

► The review focuses on high-quality RCT evidence, but inclusion of comparative observational evidence in sensitivity analyses enabled estimates for axitinib and sorafenib, which otherwise could not be connected in the network.

► The reliability of the results of this review is hampered by trial design limitations of some of the included studies: the proportional hazards assumption did not hold for PFS in the one trial including nivolumab, RCT data for axitinib and sorafenib were limited to a subgroup analysis conducted in one study, which could only be compared with the other treatments by including observational studies and the trial assessing lenvatinib with everolimus is a small phase II trial with an increased risk of a false-positive result and of over estimating the effect size due to some differences in baseline characteristics and relatively low significance level (alpha 0.15).

► There were also some differences between the trials in the network in terms of baseline characteristics, number and type of prior VEGF-targeted treatments and trial blinding, but there were too few studies to explore the potentially treatment modifying effects of these differences.

## BACKGROUND

Kidney cancers are among the most common cancers in Europe (age-standardised rates estimated at 17.2/100 000 males and 8.1/100 000 females)[1] and renal cell carcinoma (RCC) makes up 80%–90% of the new cases. RCC occurs most commonly in men older than 60 years, and smoking, obesity, hypertension, germline mutations and advanced kidney disease are established risk factors.[2] RCC is often asymptomatic until later stages, so most

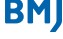

people are diagnosed with advanced or metastatic disease (amRCC); 5 year survival of amRCC is less than 10% and the goal of treatment is to slow disease progression and treat the symptoms.[2]

Targeted treatments are designed to interrupt the biological pathways needed for the cancer to grow and spread. Since 2006, eight targeted treatments have been approved by the European Medicines Agency (EMA) for the treatment of amRCC,[3–10] falling within three classes: mammalian target of rapamycin inhibitors (mTORis; everolimus[5]), tyrosine kinase inhibitors (TKIs; sunitinib,[3] pazopanib,[7] axitinib,[6] cabozantinib,[8] lenvatinib[9] (in combination with the mTORi everolimus) and sorafenib[4]), and PD-1 monoclonal antibodies (nivolumab[10]). The mechanism of action of each treatment affects tolerability and has implications for treatment choice based on patient characteristics.[11]

The emergence of targeted treatments has changed the RCC treatment pathway substantially and targeted treatments have virtually replaced the use of cytokines in many European health systems.[12] As a result, published studies assessing second-line targeted agents in populations who received first-line cytokines, or indeed adjusted indirect comparisons combining studies that enrolled those having received prior cytokines, have limited applicability to current practice. Sunitinib and pazopanib (VEGF-targeted therapies) are the only recommended first-line treatments in the latest RCC European Society for Medical Oncology (ESMO) clinical practice guidelines.[12] ESMO recommends axitinib, cabozantinib, sorafenib, everolimus, nivolumab and lenvatinib with everolimus as treatment options for second line.[12]

Second-line practice patterns are not well established, partly because some treatments have only relatively recently been approved by the EMA.[13–15] Randomised controlled trials (RCTs), cohorts and patient registry data are emerging, but head-to-head comparisons remain limited. Given the high cost of RCTs, and the number of treatments available for use at second line, it is unlikely that every treatment will ever be compared with every other treatment available. As such, adjusted indirect treatment comparisons are required to provide estimates beyond trial comparators to help establish an evidence-based treatment sequence for amRCC. Before cabozantinib, nivolumab and lenvatinib with everolimus were approved, network meta-analyses (NMAs) of RCTs or good quality observational cohorts favoured axitinib and everolimus over sorafenib, though primarily within populations who had received prior cytokines.[16–19] Two NMAs of RCTs comparing more recently approved drugs indicate that lenvatinib with everolimus or cabozantinib are likely to be the most effective option to extend overall survival (OS) and progression-free survival (PFS) in amRCC. However, neither study included all the relevant treatments and both NMAs combine evidence for people who had either received prior cytokines or VEGF-targeted agents, reflecting an outdated pathway and unreliable results given that type of prior treatment is a potential treatment effect modifier.[20]

This systematic review is the first to include randomised and observational evidence for all recently approved targeted treatments for amRCC, focusing specifically on the relevant population who have previously received a VEGF-targeted treatment. By doing so, the review aims to provide a full and clinically relevant assessment of treatment safety and clinical effectiveness, focusing on outcomes that are the most important to patients (OS, PFS, objective response rate (ORR), quality of life and adverse events).

## OBJECTIVE

To compare the safety and clinical effectiveness of targeted treatments for amRCC previously treated with VEGF-targeted therapy.

## METHODS

Methods for the review are reported in more detail in the published protocol (CRD42017071540) and were based on the principles published by the National Health Service Centre for Reviews and Dissemination.[21] The review reported here is an update and extension of a project commissioned by the UK National Institute for Health Research, registered as CRD42016042384. This review was reregistered and updated to make the results applicable outside the UK and to include treatments that have received European marketing authorisation subsequent to publication of the first iteration of the review.

### Patient and public involvement

Patients were not directly involved in the development of this review update but the original review was based on a scope produced by the National Institute for Health and Care Excellence (NICE) within which patients and patient groups were registered stakeholders.

### Eligibility criteria
#### Study design

RCTs formed the basis of the primary analyses for all outcomes. As per the published protocol, comparative observational studies were included in sensitivity analyses for OS and PFS to provide a connected network for all interventions of interest. Preclinical studies, animal studies, narrative reviews, editorials, opinions and case reports were not eligible.

#### Population

Adults (18+years) with a diagnosis of amRCC who had received previous treatment with a VEGF-targeted treatment.

#### Interventions

Interventions of interest were axitinib, cabozantinib, everolimus, lenvatinib with everolimus, nivolumab and sorafenib. Studies were included if they compared any of the listed interventions with each other, placebo or best supportive care (BSC). For the purposes of this review,

placebo was assumed to be the equivalent of BSC. Studies comparing an intervention of interest with another treatment were only included if there were insufficient direct comparisons to provide a connected network that included all treatments of interest.

## Outcomes

The primary outcomes were OS and PFS. Secondary outcomes were predefined as objective response rate (ORR), adverse events of grade 3 and above (as defined by the Common Terminology Criteria for Adverse Events) and health-related quality of life (HRQoL).

Studies were excluded if none of the outcomes of interest were reported. Comparative observational studies were only included if they reported OS or PFS in a way that could be incorporated into the NMA (ie, as a HR or where a HR could be estimated from a Kaplan-Meier curve with the number of people at risk).

## Search and selection process

Electronic searches for the original project were run in January 2016 (for RCTs; MEDLINE, EMBASE and CENTRAL) and June 2016 (observational studies; MEDLINE and EMBASE) and subsequently extended to cover a new intervention (lenvatinib with everolimus) and updated to January 2018. Manual searches of conference proceedings and bibliographies of included studies and systematic review were also updated to January 2018. Searches combined terms for the interventions of interest with condition terms for RCC and the relevant design filter (RCT or observational; example strategy provided in the online supplementary table 1). No date or language restrictions were applied. Searches for observational evidence were limited to interventions required to connect the network of treatments.

Unpublished and ongoing studies were identified by contacting experts in the field and searching Clinical-Trials.gov and the EU Clinical Trials Register.

Two reviewers screened all titles and abstracts independently. Full texts were retrieved and reviewed for records identified as potentially relevant by one or both reviewers. Discrepancies were resolved by consensus or by involving a third reviewer.

## Data extraction and quality assessment

Data extraction was carried out independently by two reviewers and cross-checked for accuracy; as with study selection, discrepancies were resolved by discussion or by involving a third reviewer. A standard data extraction form was piloted and used to capture information about study conduct, population, interventions, outcomes and risk of bias from each study, including the information source where more than one was available for a given study (template available in the online supplementary table 2 together with extracted datasets for all outcomes). Where there were incomplete information study authors were contacted to gain further details.

Methodological quality was assessed independently by two reviewers using the Cochrane Risk of Bias tool for RCTs[22] and the Risk Of Bias In Non-randomized Studies - of Interventions (ROBINS-I) for comparative observational studies.[23] Where appropriate, risk of bias was assessed separately for each outcome within a study. Disagreements were resolved by consensus or by involving a third reviewer. The likely direction and magnitude of bias across the evidence as a whole was considered during interpretation of the results.

## Data synthesis

Baseline characteristics of the included studies were compared to assess similarity of the study populations before combining results in an NMA. Fixed effects and random effects models were explored. However, as typically only one trial informed each pair-wise comparison, and hence there were little data to inform the between trial heterogeneity, a pragmatic decision was made to use the fixed effects model for all outcomes. Statistical heterogeneity was assessed using the $I^2$ statistic for pairwise comparisons and deviance information criterion for NMA. Inconsistency between direct and indirect effect estimates was assessed in closed loops in the network. Implications of observed clinical and statistical heterogeneity and inconsistency are described in the results.

Where NMA was possible, it was conducted according to the guidance described in the NICE Decisions Support Unit's Technical Support Documents for Evidence Synthesis.[24] A Bayesian Markov chain Monte Carlo approach was taken in WinBUGS v.1.4.3 software[25] (codes included in the online supplementary file) implementing uninformed priors and a burn-in of 30 000 iterations). Everolimus was specified as the baseline treatment. Data from multi-arm studies were adjusted to account for correlations in relative treatment effects.[26] OS and PFS were analysed as HRs, and adverse events and ORR were analysed as odds ratios (ORs) using participants as the unit of analysis; no formal analysis could be performed for HRQoL due to between-study variation in reporting. A 95% credible interval (CrI) can be interpreted as a 95% probability that the parameter falls within this range. If a 95% CrI doesn't include one this can, therefore, be interpreted as a statistically significant result (at the 5% level of significance). Primary analyses were based on studies of low, unclear or moderate risk of bias. Sensitivity analyses were planned for OS and PFS including RCTs of high risk of bias and observational studies of serious risk of bias. Observational studies at critical risk of bias were excluded from all analyses.

## RESULTS
### Results of the searches

Results of the original and update search and selection process are shown in figure 1.

The searches carried out in June 2016 led to the inclusion of 44 records relating to 12 studies. Five of these

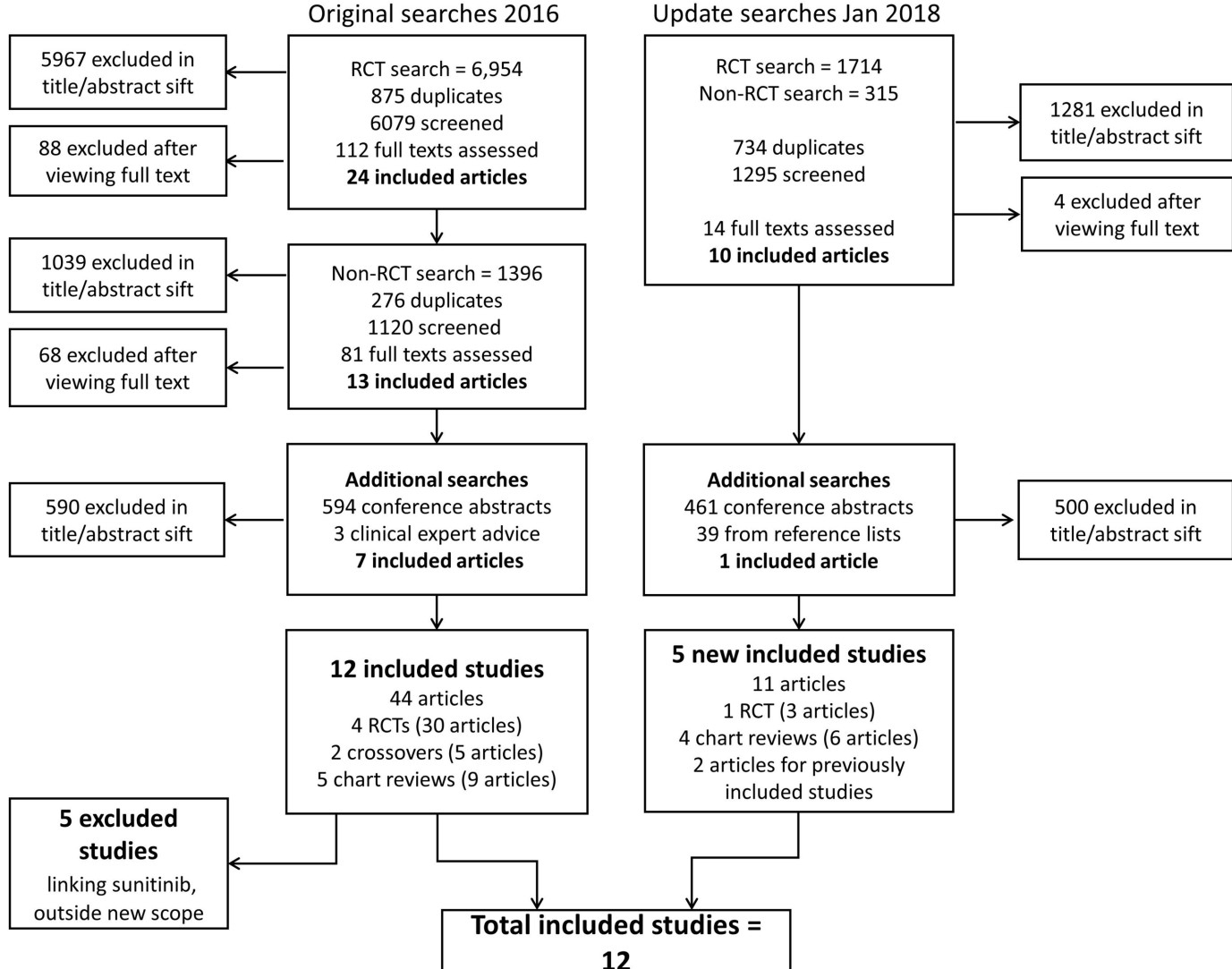

Original searches 2016

5967 excluded in title/abstract sift

RCT search = 6,954
875 duplicates
6079 screened
112 full texts assessed
**24 included articles**

88 excluded after viewing full text

1039 excluded in title/abstract sift

Non-RCT search = 1396
276 duplicates
1120 screened
81 full texts assessed
**13 included articles**

68 excluded after viewing full text

590 excluded in title/abstract sift

**Additional searches**
594 conference abstracts
3 clinical expert advice
**7 included articles**

**12 included studies**
44 articles
4 RCTs (30 articles)
2 crossovers (5 articles)
5 chart reviews (9 articles)

**5 excluded studies**
linking sunitinib, outside new scope

Update searches Jan 2018

RCT search = 1714
Non-RCT search = 315

734 duplicates
1295 screened

14 full texts assessed
**10 included articles**

1281 excluded in title/abstract sift

4 excluded after viewing full text

**Additional searches**
461 conference abstracts
39 from reference lists
**1 included article**

500 excluded in title/abstract sift

**5 new included studies**
11 articles
1 RCT (3 articles)
4 chart reviews (6 articles)
2 articles for previously included studies

**Total included studies = 12**

**Figure 1** Preferred Reporting Items for Systematic Review and Meta-Analysis diagram. RCT, randomised controlled trials.

studies have been excluded from this review because of the update of the scope excluding sunitinib as it is not recommended at second line in the most up-to-date ESMO guidance for RCC.[12] Five new studies, one RCT and four retrospective chart reviews were identified in the update and extension searches (including terms for lenvatinib with everolimus) run in January 2018, making a total of 12 included studies.[13–15 19 20 27–33]

### Included studies

Twelve studies (n=5144) met the inclusion criteria (table 1): five RCTs (one double-blind[28] and four open-label[13 15 20 28]) and seven observational studies[19 27 29–33] (retrospective cohort studies). Sample sizes varied from 101 (HOPE 205)[15] to 821 (CheckMate 025)[14] participants.

All studies recruited adults with amRCC who had received at least one prior VEGF-targeted treatment. AXIS[20] also included people who had not received prior anti-VEGF treatment, but OS and PFS data were available for the subset who had. In eight of the included studies, people had only received one prior VEGF-targeted

treatment[14 19 29–34]; the remaining five studies allowed one or more prior treatments.[13 14 27 35] Populations were predominantly male and Caucasian, and mean age was generally between 55 and 65 years. Where reported, most people had stage 3 or stage 4 clear-cell RCC and the majority had baseline ECOG performance status of 0 or 1. Baseline characteristics were generally well balanced between treatment groups within trials, with the exception of HOPE 205,[14] in which there were some imbalances in baseline characteristics, which may favour lenvatinib with everolimus over everolimus.

Where dose was reported, it was started at the standard licensed dose and adjusted according to clinical judgement. Treatment was reported in the RCTs to be continued until disease progression, unacceptable toxicity or withdrawal of consent, except for METEOR[13] and CheckMate 025[14] in which people could be treated beyond progression. Median treatment duration in the five studies where it was reported varied from 1.9 months (placebo (BSC) group of RECORD-1[28]) to 8.3 months

**Table 1** Study characteristics

| Study | Design | Location, funding | Prior treatments | Intervention | N | Type | Median age years | Male % | ECOG 0/1% | Treatment duration (follow-up) months |
|---|---|---|---|---|---|---|---|---|---|---|
| AXIS[20] | PIII OL RCT | 175 sites in 22 countries, Pfizer | One prior systemic treatment (sunitinib, cytokine or other), prior sunitinib subgroup 54% | Axitinib | 361 | CC | 61 | 73 | 99 | 8.2 (NR) |
| | | | | Sorafenib | 362 | | 61 | 71 | 100 | 5.2 (NR) |
| CheckMate 025[14] | PIII OL RCT | 146 sites in 24 countries, BMS | One or two prior targeted treatments (TKI or other, no mTORi) | Nivolumab | 410 | CC | 62 | 77 | NR | 5.5 (NR) |
| | | | | Everolimus | 411 | | 62 | 74 | | 3.7 (NR) |
| HOPE 205[15] | PII OL RCT | 37 sites in Czech Republic, Poland, Spain, UK, US, Eisai | one prior TKI, no prior mTORi | Lenvatinib+eve | 51 | CC | 61 | 69 | 100 | 7.6 (NR) |
| | | | | Everolimus | 50 | | 59 | 76 | 100 | 4.1 (NR) |
| METEOR[13] | PIII OL RCT | 173 sites in 26 countries, Exelixis | one or more prior TKIs; no prior mTORi | Cabozantinib | 330 | CC | 63* | 77 | 100 | 8.3 (18.7) |
| | | | | Everolimus | 328 | | 62* | 73 | 100 | 4.4 (18.8) |
| RECORD-1[28] | PIII DB RCT, Novartis | 86 sites in Australia, Canada Europe, Japan, US, Novartis | one or two prior TKIs; no prior mTORi | Everolimus | 277 | CC | 61* | 78 | NR | 4.6 (NR) |
| | | | | BSC/placebo | 139 | | 60* | 76 | | 1.9 (NR) |
| Guida 2017[32] | Chart review | One site in France, NR | One prior targeted treatment (TKI or other) | Everolimus | 81 | 92% CC | 57 | 69 | 85 | NR (33) |
| | | | | Axitinib | 45 | | 54 | 78 | 82 | NR (26) |
| Heng 2016[19] | Chart review | UK, Germany, France, Netherlands, Novartis | One prior TKI (sunitinib or pazopanib) | Everolimus | 115 | NR | 60.2 | 66.7 | 91.8%≤2 | NR (NR) |
| | | | | Axitinib | 96 | | | | | NR (NR) |
| | | | | Sorafenib | 98 | | | | | NR (NR) |
| Iacovelli 2015[27] | Chart review | 23 sites in Italy, NR | Two prior targeted treatments (TKI or other) | Sorafenib | 90 | CC | 63 | 74 | 81 | NR (NR) |
| | | | | Everolimus | 143 | | | | | NR (NR) |
| Lakomy 2017[23] | Chart review | Czech national registry, † | one prior targeted treatment (TKI or other) | Everolimus | 520 | 94% CC | 65 | 75 | 95 | 6.1 (NR) |
| | | | | Sorafenib | 240 | | 62 | 75 | 90 | 7.1 (NR) |
| SPAZO-2[31] | Chart review | 50 sites in Spain, Novartis | one prior TKI (pazopanib) | Everolimus | 101 | 88% CC | 66 | 64 | NR | NR (28) |
| | | | | Axitinib | 88 | | 63 | 68 | | |
| Vogelzang 2016[29] | Chart review | US, Novartis | One prior TKI; no prior cytokines | Everolimus | 325 | 85% CC | 61* | 70 | 80 | NR (15*) |
| | | | | Axitinib | 127 | | 60* | 65 | 84 | NR (13*) |
| Wong 2014[30] | Chart review | US, Novartis | One prior TKI; no prior mTORi, cytokines, bevacizumab | Everolimus | 233 | 91% CC | 64 | 70 | NR | NR (12.9) |
| | | | | Sorafenib | 123 | | 66 | 72 | | NR (12.1) |

*Mean values where median was not reported.
†Ministry of Health of the Czech Republic, Central European Institute of Technology, The Ministry of Education, Youth and Sports. RENIS registry part funded by Pfizer, Bayer and Novartis.
amRCC, advanced or metastatic RCC; BSC, best supportive care; cc, clear cell variant; CC, clear cell variant; mRCC, metastatic renal cell carcinoma; mTORi, mammalian target of rapamycin inhibitor; ncc, non-clear cell variant; NR, not reported; RCT, randomised controlled trials; TKI, tyrosine kinase inhibitor.
Notes: ECOG percentages that do not total 100 are due to missing data.

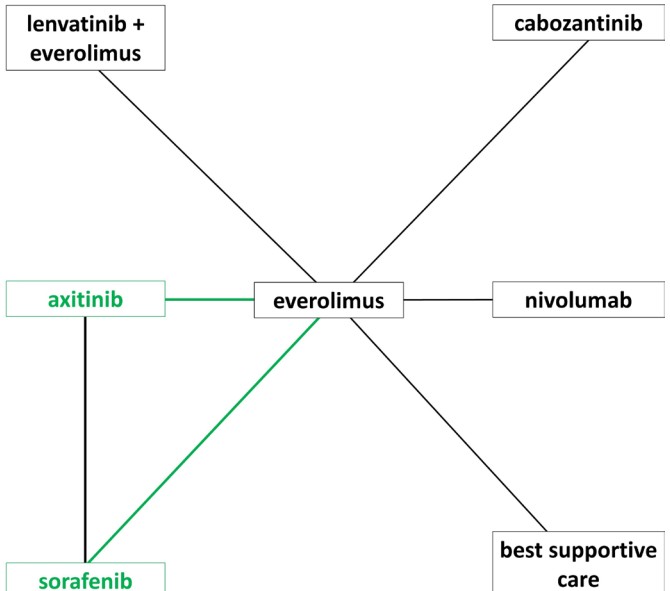

**Figure 2** Network diagram. Direct comparisons made by RCTs are shown by black lines, and the additional connections possible by incorporating comparative observational studies are shown with green lines; axitinib and sorafenib did not connect to the other treatments using only RCT evidence.

(cabozantinib group of METEOR).[13] Median length of follow-up ranged from 12.1 months to 23.6 months, but was only reported in four studies.

Most studies gave limited information regarding treatments received subsequent to the study drug. In RECORD-1,[28] 76% of people randomised to placebo received open-label everolimus at progression, but the confounding of OS was reduced by using crossover-adjusted data in the NMA. Treatment crossover was not reported to have occurred in any other studies.

Treatments compared in each of the studies are shown in table 1 and figure 2. Nivolumab could not be connected in the PFS network because it was not appropriate to analyse CheckMate 025[36] data with a Cox proportional hazards model.

### Risk of bias
The five RCTs[13 15 20 28 36] were of good methodological quality; all are at low risk of bias for random sequence generation and allocation concealment. RECORD-1[28] was the only blinded study so there is a risk of performance bias in the others. In general, OS and PFS are considered low risk of detection and reporting biases for all RCTs except for a high risk of PFS detection bias in CheckMate 025[14] because it was not assessed by an independent review committee. None of the outcomes in the RCTs were at high risk of attrition bias; all used appropriate censoring for the time-to-event analyses, although OS data from CheckMate 025[14] and METEOR[13] are immature. Other possible sources of bias pertain to group differences in the rate and type of subsequent treatments received, which were poorly reported in most trials.

RECORD-1[28] was the only trial allowing cross-over for people in the placebo arm, although cross-over adjusted results were reported. Despite appropriate randomisation in HOPE 205,[15] which is a small phase II trial, there were some imbalances in the baseline characteristics of the people in the trial, which may indicate a better prognosis for the lenvatinib with everolimus group compared with everolimus alone. In addition, alpha was set to 0.15, compared with the usual 0.05, and HOPE 205 is therefore of a higher risk of a false-positive result and possibly of over-estimating the effect size.

The observational studies included in the OS and PFS sensitivity analyses are at a higher risk of bias than the RCTs. Overall ROBINS-I ratings were at best moderate, for OS,[19 29 31] and serious risk of bias for PFS. One study was at critical risk of bias for both PFS and OS,[33] which was excluded from the sensitivity analyses. In all studies the potential for inadequate control for confounding was thought to increase the risk of bias. All studies reporting PFS also had an increased risk of bias for this outcome due to the lack of standardised measurement for assessing progression and that outcome assessors were aware of the interventions.

One of the observational studies was publicly funded,[33] two studies did not report their funding source[27 32] and the remaining observational studies and all RCTs were sponsored by various pharmaceutical companies. Risk of bias assessments for all included studies are provided in the online supplementary tables 3 and 4.

### Overall survival
Lenvatinib with everolimus, cabozantinib and nivolumab all showed statistically significant benefits over the baseline treatment, everolimus, in in the primary OS analysis (table 2). Lenvatinib with everolimus had the highest probability (61%) of being the most effective treatment out of those compared in the primary analysis. These results were mirrored in the sensitivity analysis including observational studies. The sensitivity analysis also suggests everolimus may be more effective than axitinib, sorafenib and BSC for overall survival. However, there is evidence of inconsistency between the direct and indirect evidence for axitinib, sorafenib and everolimus, which indicates that there is heterogeneity between the studies and highlights the uncertainty around the true estimates of the relative effect of these treatments. Raw data for OS and all other outcomes are available in the online supplementary tables 5 to 9.

### Progression-free survival
As with OS, lenvatinib with everolimus and cabozantinib both showed statistically significant benefits over everolimus, and lenvatinib with everolimus had the highest probability (66.5%) of being the most effective treatment out of those compared in the primary analysis of PFS (table 2). The results of the sensitivity analysis including observational study data indicate that axitinib also improves PFS compared with everolimus, whereas BSC

 Karner C, *et al. BMJ Open* 2019;**9**:e024691. doi:10.1136/bmjopen-2018-024691

**Table 2** Results of the network meta-analyses for the primary outcomes (OS and PFS) and grade 3 or grade 4 adverse events

| | Primary NMA of RCTs | | Sensitivity NMA of RCTs and observational studies |
|---|---|---|---|
| Overall survival | Probability most effective (%) | HR *versus* everolimus (95% credible interval) | |
| Lenvatinib+everolimus | 61 | **0.61 (0.36 to 0.96)** | **0.61 (0.36 to 0.96)** |
| Cabozantinib | 28 | **0.66 (0.53 to 0.82)** | **0.66 (0.53 to 0.83)** |
| Nivolumab | 10 | **0.74 (0.57 to 0.93)** | **0.74 (0.57 to 0.93)** |
| Axitinib | – | – | 1.14 (0.95 to 1.37) |
| Sorafenib | – | – | 1.38 (1.12 to 1.68) |
| BSC | 2 | 1.90 (0.61 to 4.53) | 1.90 (0.60 to 4.56) |
| Progression-free survival | Probability most effective (%) | HR *versus* everolimus (95% credible interval) | |
| Lenvatinib+everolimus | 67 | **0.47 (0.26 to 0.77)** | **0.47 (0.26 to 0.77)** |
| Cabozantinib | 34 | **0.51 (0.41 to 0.63)** | **0.51 (0.41 to 0.63)** |
| Axitinib | – | – | 0.84 (0.70 to 1.00) |
| Sorafenib | – | – | 1.17 (0.95 to 1.43) |
| BSC | 0 | **3.06 (2.31 to 3.97)** | **3.06 (2.31 to 3.97)** |
| Grade 3 or four adverse events | Probability least harmful (%) | OR *versus* everolimus (95% credible interval) | |
| Lenvatinib+everolimus | 0 | **2.67 (1.05 to 5.68)** | – |
| Cabozantinib | 0 | **1.66 (1.18 to 2.27)** | – |
| Nivolumab | 100 | 0.40 (0.29 to 0.55) | – |

SC, best supportive care; NMA, network meta-analysis; OS, overall survival; PFS, progression-free survival; RCT, randomised controlled trial. Numbers in bold are statistically significant.

leads to significantly shorter PFS compared with everolimus, and there was no statistically significant difference between everolimus and sorafenib. For PFS, there was no evidence of inconsistency between the direct and indirect evidence of axitinib, sorafenib and everolimus.

Nivolumab was not included in the analyses of PFS because the proportional hazards assumption does not hold for this outcome in CheckMate 025.[14]

### Objective response rate

Two of the four RCTs that could be included in the NMA for ORR observed no events in one treatment arm (everolimus in HOPE 205[15] and BSC in RECORD-1),[28 35] causing the results from the NMA to be unreliable and lack face validity. Results using a 0.5 correction for 0 values indicate that treatment with cabozantinib, lenvatinib with everolimus and nivolumab all lead to a better response rates than treatment with everolimus, which in turn in significantly better than BSC (online supplementary table 10).

### Adverse effects

In terms of safety, nivolumab had the highest probability of being least harmful, that is, the rate of grade 3 or grade 4 adverse events (AEs) was significantly lower with nivolumab (18.7%) than with everolimus (36.5%),[14] whereas treatment with either cabozantinib or lenvatinib with everolimus resulted in significantly higher rates of grade 3 or grade 4 AEs than everolimus (METEOR[13]: cabozantinib 71.0%, everolimus 59.9%; HOPE 205[15]: lenvatinib +everolimus 71%, everolimus 50%). Rates of

grade 3 or grade 4 AEs were not reported for axitinib or BSC in AXIS and RECORD-1.[34 35]

### Health-related quality of life

Treatments could not be compared using NMA for HRQoL as different measures and tools were used for assessments. HRQoL scores were similar between axitinib and sorafenib in AXIS[34] and results favoured nivolumab over everolimus in CheckMate 025.[14] Results in RECORD-1[28] favoured BSC over everolimus, although this effect was only apparent if models were used to account for data not missing at random. METEOR[13] results were similar for everolimus and cabozantinib. HRQoL was not measured in HOPE 205.[15] A summary of results from each of the five RCTs is provided in the online supplementary file.

### DISCUSSION

This systematic review and network meta-analysis suggests that lenvatinib with everolimus, cabozantinib and nivolumab all prolong PFS and are likely to increase OS compared with everolimus for people with amRCC previously treated with VEGF-targeted therapy. The results suggest lenvatinib with everolimus is likely to be the most effective treatment, followed by cabozantinib and then nivolumab, but there is considerable uncertainty around how they compare to each other and how much better they are than the earlier generation of targeted treatments, axitinib and sorafenib. Nivolumab may be

associated with fewer grade 3 or grade 4 AEs than treatment with both lenvatinib with everolimus and cabozantinib. All treatments considered in this review appear to delay disease progression and prolong survival more than providing BSC, and results for ORR support the primary OS and PFS analyses. Due to differences in reporting and HRQoL tools used, it was not possible to perform NMAs on HRQoL.

This is a robust and comprehensive systematic review and NMA based on the principles published by Centre for Reviews and Dissemination[21] using the Meta-analysis Of Observational Studies in Epidemiology (MOOSE[37]) and Preferred Reporting Items for Systematic Reviews and Meta-Analyses[38] reporting guidelines, and conducted according to prespecified methods in a prospectively registered protocol (PROSPERO CRD42017071540). The inclusion of all recently approved treatments increases the relevance and timeliness of the review. The review is also highly relevant as it focuses on the effectiveness and safety of these treatments when used after first line TKI treatment, as recommended in clinical guidelines.[12] However, there is not enough evidence available to answer questions about the sequencing of later lines of treatments.

Although this study focuses on high-quality RCT evidence, the inclusion criteria were widened to incorporate comparative observational evidence in sensitivity analyses to enable estimates for axitinib and sorafenib, which otherwise could not be connected to the network.

However, the robustness of the evidence in this review is limited by several factors:

1. PFS for nivolumab compared with the other treatments could not be estimated in this review because the proportional hazards assumption didn't hold for this outcome in the one trial including nivolumab.[14]
2. Relevant RCT data for axitinib and sorafenib were limited to a subgroup analysis conducted in one study that did not connect to the network of other RCTs.[34] Axitinib and sorafenib could only be compared with the other treatment options by including observational studies, which were generally at a serious risk of different kinds of bias.
3. The trial assessing the efficacy of lenvatinib with everolimus is a small phase II trial, with an alpha set to 0.15 and therefore a higher than usual risk of false-positive results and overestimation of the treatment effect. In this trial there were also some differences in baseline characteristics likely to lead to an over-estimation of the treatment effect of lenvatinib and everolimus compared with everolimus, which introduces uncertainty around the true treatment effect.
4. Although the baseline characteristics were well balanced within most of the trials, there were some differences in performance status and number of prior VEGF-targeted treatments between the trials. There were also differences in trial design with some trials being double-blind or open label. Outcome assessment was not always done by an independent review committee. However, in the nivolumab trial, CheckMate 025,[14] progression was only assessed by non-blinded trial investigator. There were too few studies to explore the effects of these differences between studies, which is a limitation and increases the uncertainty of the results.
5. The number of studies identified prevented meaningful subgroup analyses to explore potentially important prognostic factors that varied across the included studies. For example, while the review was limited to populations who had received prior VEGF therapy, there was variation in eligibility and baseline criteria regarding the type of VEGF treatment received and number of prior lines (see table 1).

Two NMAs of different subsets of treatments for previously treated amRCC have recently been published.[39 40] Unlike these studies, this review provides an alternative approach and a comparison between all recently approved treatments. Rassy et al[40] and Amzal et al[39] combine evidence for people who had either received prior cytokines or VEGF-targeted agents. This enabled a connected network using only RCT data but the type of prior treatment has been shown to be a potential treatment effect modifier,[34] which could introduce bias into the analysis. In addition, results for people who have only had prior cytokines are less relevant to clinical practice than for prior VEGF-targeted treatments as most people receive a TKI first line, in line with clinical guidelines.[12] The NMAs of Amzal et al[39] and Rassy et al[40] are also limited by the reliance on the Treatment Approaches in Renal Cancer Global Evaluation Trial (TARGET) trial[41] to link axitinib and sorafenib to the network analysed. TARGET[41] is an RCT of sorafenib and placebo in which people only had prior cytokines and not prior TKI. The results from the TARGET trial are also confounded by crossover, which has only been partly accounted for by using immature data censored at crossover, and the lack of proportional hazards between the trial arms for PFS and OS.

For the trials that are shared between Amzal et al[39] and this review and Rassy et al[40] and this review the order of treatments, in terms of OS and PFS, is similar. However, this systematic review focuses specifically on the most relevant population, who have previously received a VEGF-targeted treatment, and avoids the issues with the TARGET[41] trial by including both randomised and observational evidence, and thereby provides more relevant and reliable estimates of the relative efficacy between all the interventions.

Neither prior review planned to assessed ORR or HRQoL and so these outcomes cannot be compared with previous results. A narrative presentation of adverse events in Rassy et al[40] is in line with our findings that lenvatinib with everolimus is likely to be less well tolerated than nivolumab; Rassy and colleagues highlight that similarly high proportions of patients experienced Grade 3–4 adverse events and discontinued treatment due to toxicity on cabozantinib and lenvatinib with everolimus, and the most commonly reported tolerability issues across treatments were fatigue and diarrhoea.

All treatments considered in this review delay disease progression and prolong survival more than BSC, and although this review gives an indication of the ranking of the most effective treatments for treating recurrent amRCC, there is still much uncertainty around how much these treatments differ from each other in terms of effectiveness and safety. The choice of treatment should take into account patient preference, comorbidities, symptoms, tumour burden and how aggressive the cancer is. Policy-makers also need to consider the cost-effectiveness of the treatments.

It would be preferable to have high-quality RCT data comparing all the available RCC treatment options, but this is unlikely to be commissioned due to the high costs of clinical trials. However, what is more likely and still needed is a larger RCT of lenvatanib with everolimus to confirm the efficacy data from the current phase II trial with its small sample size. RCT data of axitinib and sorafenib versus other comparators in the network are also required to enable higher quality evidence for these comparisons. As there is no cure for amRCC and as virtually all people progress, research is needed into the development of resistance to treatments. Further research is also required into the impact of different sequencing of drugs from second line and onwards as more people are well enough to tolerate additional lines of treatment and most of these drugs are approved for use also beyond second line (cabozantinib, everolimus, and nivolumab).

**Acknowledgements** The authors would like to thank Clare Fiatikoski (BMJ) and George Osei-Assibey (BMJ) for their contribution to the original literature searching and data extraction underpinning this review, and Samantha Barton (BMJ) for providing comments on the draft manuscript. The authors would also like to thank Dr Amit Bahl (University Hospitals Bristol NHS Foundation Trust, Bristol), Dr. Lisa Pickering (St Georges Hospital, London), Dr. Sarah Rudman (Guys and St Thomas Hospital, London), Dr. Penny Kechagioglou (University Hospitals Coventry and Warwickshire, Coventry), Professor Martin Gore (Royal Marsden NHS Foundation Trust, London) and Dr. Naveed Sarwar (Imperial College Healthcare NHS Trust, London) for providing clinical advice throughout the original project.

**Contributors** CK validated data extraction, carried out and validated meta-analyses, and drafted and edited the manuscript. KK carried out additional searches, contributed to the appraisal of title and abstracts, assessment of full publications for inclusion, data extraction and validation, carried out meta-analyses and drafted the manuscript. VW devised and carried out database searches, contributed to the appraisal of title and abstracts, assessment of full publications for inclusion, and data extraction. NM contributed to the appraisal of title and abstracts, the assessment of full publications for inclusion and data extraction. SJE supervised the production of the manuscript and acted as methodological adviser.

**Funding** The review presents an update and extension to independent research funded by the UK National Institute for Health Research (NIHR) health technology assessment programme (HTA 16/58/01).

**Disclaimer** The views and opinions expressed in this report are those of the authors and not necessarily those of the NHS, the NIHR or the Department of Health and Social Care.

**Competing interests** None declared.

**Patient consent for publication** Not required.

**Provenance and peer review** Not commissioned; externally peer reviewed.

**Data sharing statement** Search strategies, data extraction form, risk of bias summaries, data inputs, NMA code and results tables are provided in a online supplementary file.

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
