## [Reviewer comments · BMJ Open]

This paper was submitted to a another journal from BMJ but declined for publication following peer review. The authors addressed the reviewers' comments and submitted the revised paper to BMJ Open. The paper was subsequently accepted for publication at BMJ Open.

(This paper received three reviews from its previous journal but only two reviewers agreed to published their review.)

ARTICLE DETAILS

TITLE (PROVISIONAL)	TARGETED THERAPIES FOR PREVIOUSLY TREATED ADVANCED OR METASTATIC RENAL CELL CARCINOMA: SYSTEMATIC REVIEW AND NETWORK META-ANALYSIS
AUTHORS	Karner, Charlotta; Kew, Kayleigh; Wakefield, Victoria; Masento, Natalie; Edwards, S

VERSION 1 – REVIEW

REVIEWER	Janice P. Dutcher, M.D. Associate Director, Cancer Research Foundation, INC, of New York. USA
REVIEW RETURNED	02-Aug-2018

GENERAL COMMENTS	1. This manuscript is a systematic review and network meta-analysis of studies of “targeted” therapies for advanced or metastatic renal cell cancer. These include both randomized controlled trials and observational studies as well as smaller studies.2. The authors describe the strength as being an evaluation only of subjects with prior anti-VEGF therapies, not prior cytokines, which is important. They describe the used of mixed studies for meta-analyses as a flaw in references 32 and 33 since outcomes by different prior therapies has shown a different effect from subsequent anti-VEGF therapy. Nevertheless, the type and AMOUNT of prior therapy may impact outcome of trials, and they do not separate out those with only one prior anti-VEGF therapy, versus more than one. Can this be done?3. The authors carefully outline the limitations of the analysis on page 21, which I think are quite critical to evaluating the current status of RCC treatment, particularly if attempting to create a hierarchy. I suggest that these should be given greater emphasis, particularly, # 4. Patient characteristics often influence treatment selection, even in clinical trials.4. I suggest that the paragraph following the limitations should be the first paragraph of the discussion, rather than trying to “pick a winner” based on the analysis provided. The first sentence of the discussion overstates the ability to define a “winner” based on cross-study evaluations. This is my major recommendation.5. This is particularly the case, since different PFS and OS lists, and not using ORR or QOL.
---

	6. The conclusion of the abstract should also be softened based on the limitations. 7. It is also important to note different mechanisms of action of the therapies evaluated, because this may also direct treatment decisions. Grouping all the anti-VEGF agents as TKIs, does not distinguish potential other targets, especially with cabozantinib. (bkgnd section) 8. Similarly, Nivolumab should not be characterized as simply a monoclonal antibody (bkgnd line 41) but should be characterized by its target, PD-1, and therefore its effect is through inhibition of the target. 9. The point of 7 and 8 is that these differences may also lead to differences in patient selection for these treatments based on biomarkers. Minor points: a few spelling errors: pg 6, line 29, treatment; pg 21, line 26, axitinib.
--	--

REVIEWER	Daniel Spratt University of Michigan USA
REVIEW RETURNED	04-Sep-2018

GENERAL COMMENTS	This is a very well performed network meta-analysis investigating optimal targeted therapies in RCC treated patients. The bayesian approach allows for probabilities to be generated which are highly informative and a welcomed method to inform what an optimal treatment may be without conducting a dedicated RCT (for which there are too many combination now to conduct RCTs for every comparison). This is a rare time for which I do not have any substantive comments and I believe the authors have performed a rigorous study, quantified limitations and potential biases, and stated appropriate conclusions supported by the data.
---

REVIEWER	Celia Álvarez Bueno Castilla-La Mancha University, Spain
REVIEW RETURNED	29-Oct-2018

GENERAL COMMENTS	Thank you for the opportunity to review this interesting paper that provides a comprehensive review on an important health related issue. However, there are some concerns that should be addressed in order to clarify the aims and methods of this paper. INTRODUCTION First and second paragraphs should be joined and organized from general information to more specific one. The fifth paragraph should be limited to the information in other research papers, all the information regarding the treatment options should be stated in the previous paragraph. The objective in the last paragraph does not match with the one in objective section and with the first paragraph in the discussion section. Please, unify in order to provide a clear aim to the reader. There are complete paragraphs without references. This should me amended.
--

	METHODS/RESULTS There are secondary outcomes included from which there is no information in the introduction section. If these outcomes are important to understand the picture, then they and their role in the research should be stated in the introduction section. Search strategy is not update. Please, update it, nine months are too much for a review. The used of statistical methods should be referenced. (i.e., random or fixed effects models; heterogeneity interpretation,...) Each supplementary table should be referenced in the appropriated place of the test. Non-RCT are defined as: An experimental study in which people are allocated to different interventions using methods that are not random. It should be clarified why cohort studies and restrospective chart reviews are included. Additionally, I doubt on how including these designs could fit in a network meta-analysis. DISCUSSION The authors' study definition change across the paper, please unify it (network, comprehensive, ...) Second and third paragraph are related to strengths and limitations, these should be stated at the end of the discussion section. I miss more discussion on the results across the discussion section. Please, discuss all the outcomes of interest comparing them with previous findings.
--	---

VERSION 1 – AUTHOR RESPONSE

Reviewer 1: Janice Dutcher

1. No response required
2. We appreciate the reviewer's suggestion regarding type and amount of prior therapy, which was a subgroup analysis in the original health technology assessment. Too few studies were available to conduct reliable subgroup analyses, so they were not included in the protocol for this update. Inclusion criteria for type and amount of prior therapies in each included study is included in Table 1, and we have added the point to the discussion.
3. We have amended the opening of the discussion and abstract conclusions to better account for the uncertainties and limitations listed.
- 4–6. The first two paragraphs of the discussion have been reorganised and reworded to explain the uncertainty in the treatment hierarchy, which has also been amended in the abstract conclusions.
- 7–9. A sentence has been added to the background about targets and mechanism of action.

Reviewer 2: Daniel Spratt
No changes suggested

Reviewer 3: Celia Álvarez Bueno
The first two paragraphs have been merged and reorganised as requested.

We appreciate the reviewer's suggestion to changes of the fifth paragraph, but note that the treatment options are stated in the previous paragraph and that paragraph five is focused on other research papers available.

The wording of objectives has been unified in the background, objectives and discussion.

We have added additional references to the background.

The final sentence of the introduction has been amended to add in the relevance of all outcomes.

We appreciate the reviewer's comment about the date of the search but updating at this late stage would further delay the results being published. We are aware of emerging evidence in the field and are relatively certain that any studies published since the last search would be observational, so only the sensitivity analyses would be affected.

Statistical methods, including those requested by the reviewer (random or fixed effects models; heterogeneity interpretation) are described in the Data synthesis section.

Table numbers from the supplementary file have been added to the text.

We accept the possible confusion with the definition of non-RCT, which we intend to mean comparative evidence not from RCTs. We have amended the wording from 'non-RCTs' to observational studies throughout the manuscript. Only comparative observational studies were eligible, meaning the between-group data could be included in NMA in the same way as for the RCTs. We have unified the way the design is described to systematic review and network meta-analysis, in line with the title.

The discussion was written in accordance with the structure suggested by BMJ Open so we have not reorganised as suggested by the reviewer.

We have added a paragraph to compare results for secondary outcomes with previous reviews.

VERSION 2 – REVIEW

REVIEWER	Janice P. Dutcher Cancer Research Foundation Data safety and monitoring committees: Amgen, Merck, Eisai, BMS, Tracoon, PrECOG. Consultant: Prometheus, Nektar.
REVIEW RETURNED	09-Dec-2018

GENERAL COMMENTS	1. I am still concerned regarding the clinical applicability of these data, based on how the original studies were designed, and thus the data provided for a meta-analysis. we have subsequently learned much about RCC heterogeneity, and patients who particularly would respond to MTOR inhibitors and Anti-PD1 inhibitors - and these are subgroups, that might not be appropriate for secondary anti-VEGF. Additionally, the design of many of the "registration" trials for new anti-VEGF TKIs purposely chose a weak, or obsolete comparator, thus biasing the result. 2. Error in abstract conclusion: First generation targeted: Sorafenib, sunitinib, temsirolimus, everolims second Gen: Pazo and Axit 3rd Gen : Cabo, and combo of Lenvatinib and everolimus; Nivo is a totally different class of drugs 3.Bkground para 2 - anti-PD-1 - not patient biomarkers, but patient characteristics. 4.Again, patient and clinical characteristics may inform the type of treatment to utilize - based again on the heterogeneity that we are discovering.
--

	5. Discussion - I would suggest NOT providing a hierarchy of treatment choice - again, based on issues discussed in #4, not would I do so for toxicity - again, this depends. Some of lack of tox of Nivo is lack of response, yet if get immune toxicity - may predict for response 7. Again, would not make strong recommendations based on comment above, but also - only 1 Nivo trial, indirect comparisons, and the Levatinib/Ev trial is phase II. 8. Pg 25 - should not recommend sorafenib - old and not as good as any of the other anti-VEGF drugs. just cheap.
REVIEWER	Celia Álvarez-Bueno Universidad de Castilla-La Mancha Health and Social Research Center, Spain
REVIEW RETURNED	29-Nov-2018
GENERAL COMMENTS	The reviewer completed the checklist but made no further comments.

VERSION 2 – AUTHOR RESPONSE

1. I am still concerned regarding the clinical applicability of these data, based on how the original studies were designed, and thus the data provided for a meta-analysis. We have subsequently learned much about RCC heterogeneity, and patients who particularly would respond to MTOR inhibitors and Anti-PD1 inhibitors - and these are subgroups, that might not be appropriate for secondary anti-VEGF. Additionally, the design of many of the "registration" trials for new anti-VEGF TKIs purposely chose a weak, or obsolete comparator, thus biasing the result.

Response:

The systematic review was conducted in a methodologically robust manner, where the issues raised by the reviewer were assessed and discussed, where necessary. The trials used in the network for the network meta-analysis were considered sufficiently robust with any potential biases explicitly stated (e.g. in the subsequent analysis including observational studies). In terms of the comparators within the trials, some have limited use in clinical practice. However, this will not have an effect on the results of the network meta-analysis. It is highlighted in the review that sequencing of treatments may be very important but that there aren't data available at the moment to explore this. No change made.

2. Error in abstract conclusion: First generation targeted: Sorafenib, sunitinib, temsirolimus, everolimus
second Gen: Pazopanone and Axitinib

3rd Gen : Cabozantinib, and combo of Lenvatinib and everolimus;

Nivo is a totally different class of drugs

Response: The authors agree that there may be some ambiguity about the definition of different generations of treatments. The reference to a "first generation" has been changed to "earlier generation".

3. Background para 2 - anti-PD-1 - not patient biomarkers, but patient characteristics.

Response: The authors would like to clarify that anti-PD-1 is not referred to as a biomarker in the text, but that treatment choice may be based on biomarkers. However, for clarity the text has been changed from "patient biomarker" to "patient characteristics".

4. Again, patient and clinical characteristics may inform the type of treatment to utilize - based again on the heterogeneity that we are discovering.

Response: A comparison of the efficacy and safety of different therapies using network meta-analyses of trial data will not provide recommendations for individual patients but can inform treatment choices on a population level and as mentioned in the discussion of the manuscript, the choice of treatment for individual patients needs to take into account, e.g. patient preference, comorbidities, symptoms and prior therapy. No change made.

5. Discussion - I would suggest NOT providing a hierarchy of treatment choice - again, based on issues discussed in #4, not would I do so for toxicity - again, this depends. Some of lack of tox of Nivo is lack of response, yet if get immune toxicity - may predict for response

Response:

The authors agree that a hierarchy of treatment choice would be inappropriate when it isn't based on strong evidence. This is why we have avoided providing a ranking of individual treatments where there has been substantial uncertainty and overlapping of credible intervals. Instead we have only concluded that lenvatinib + everolimus, cabozantinib and nivolumab are all effective in prolonging survival, but there is considerable uncertainty about how they compare to each other and how much better they are than axitinib and sorafenib. It has also been emphasised in the text that nivolumab may be associated with fewer grade 3 or 4 adverse events, which is what the trial data indicates. No change made.

7. Again, would not make strong recommendations based on comment above, but also - only 1 Nivo trial, indirect comparisons, and the Levatinib/Ev trial is phase II.

Response: The limitations of the individual trials, e.g. that the trial of lenvatinib + everolimus is a phase II study with a small sample size are highlighted in the discussion. No change made.

8. Pg 25 - should not recommend sorafenib - old and not as good as any of the other anti-VEGF drugs. just cheap.

Response:

The authors agree with the reviewer that sorafenib is less effective than several of the other anti-VEGF therapies (lenvatinib+everolimus, cabozantinib, nivolumab), as indicated by the results of the review. However, we highlight the lack of RCT data between sorafenib and the newer therapies, which would be needed to confirm the results of this review. Although the acquisition cost of sorafenib is low, the cost-effectiveness of all of these treatments is unclear and would be the subject of future research. No change made.